# A Comprehensive Review of the Effects of Glycemic Carbohydrates on the Neurocognitive Functions Based on Gut Microenvironment Regulation and Glycemic Fluctuation Control

**DOI:** 10.3390/nu15245080

**Published:** 2023-12-12

**Authors:** Jian Yin, Li Cheng, Yan Hong, Zhaofeng Li, Caiming Li, Xiaofeng Ban, Ling Zhu, Zhengbiao Gu

**Affiliations:** 1School of Food Science and Technology, Jiangnan University, Wuxi 214122, China; 7200112038@stu.jiangnan.edu.cn (J.Y.); hongyan@jiangnan.edu.cn (Y.H.); zfli@jiangnan.edu.cn (Z.L.); caimingli@jiangnan.edu.cn (C.L.); banxiaofeng@jiangnan.edu.cn (X.B.); zhuling@jiangnan.edu.cn (L.Z.); 2State Key Laboratory of Food Science and Technology, Jiangnan University, Wuxi 214122, China; 3Collaborative Innovation Center of Food Safety and Quality Control in Jiangsu Province, Jiangnan University, Wuxi 214122, China; 4National Engineering Research Center for Functional Food, Jiangnan University, Wuxi 214122, China

**Keywords:** glycemic carbohydrates, digestive properties, glycemic fluctuation, gut microbiota, glucose metabolism, cognitive functions

## Abstract

Improper glycemic carbohydrates (GCs) consumption can be a potential risk factor for metabolic diseases such as obesity and diabetes, which may lead to cognitive impairment. Although several potential mechanisms have been studied, the biological relationship between carbohydrate consumption and neurocognitive impairment is still uncertain. In this review, the main effects and mechanisms of GCs’ digestive characteristics on cognitive functions are comprehensively elucidated. Additionally, healthier carbohydrate selection, a reliable research model, and future directions are discussed. Individuals in their early and late lives and patients with metabolic diseases are highly susceptible to dietary-induced cognitive impairment. It is well known that gut function is closely related to dietary patterns. Unhealthy carbohydrate diet-induced gut microenvironment disorders negatively impact cognitive functions through the gut–brain axis. Moreover, severe glycemic fluctuations, due to rapidly digestible carbohydrate consumption or metabolic diseases, can impair neurocognitive functions by disrupting glucose metabolism, dysregulating calcium homeostasis, oxidative stress, inflammatory responses, and accumulating advanced glycation end products. Unstable glycemic status can lead to more severe neurological impairment than persistent hyperglycemia. Slow-digested or resistant carbohydrates might contribute to better neurocognitive functions due to stable glycemic response and healthier gut functions than fully gelatinized starch and nutritive sugars.

## 1. Introduction

Glycemic carbohydrates (GCs), easily digested and absorbed by the human body and triggering glycemic response, are one of the most essential and economical sources of dietary energy for humans [1]. They mainly consist of sugars (glucose, fructose, sucrose, and maltose), oligosaccharides (malto-oligosaccharides, maltodextrins), and polysaccharides (starches) based on their degree of polymerization. Glucose is an essential fuel for the brain, and glycemic carbohydrate intake plays a crucial role in human evolution. The preference for GCs contributes to an increasing number of younger patients with metabolic diseases, such as obesity and diabetes [2], and exerts a negative impact on cognition, especially in children, the elderly, and diabetics. Therefore, understanding the effects and mechanisms of different carbohydrates on neurocognitive function is a prerequisite for a reasonable selection of diet and the search for therapeutic targets for related diseases.

Different GCs can lead to different digestion rates and products due to differences in molecular size, structure, and anti-enzymatic hydrolysis ability. Unabsorbed digestive products can be used as substrates to remodel the gut microbiota and microenvironment after entering the colon. The gut microbiota metabolites, such as short-chain fatty acids (SCFAs), neurotransmitters, and lipopolysaccharides (LPS), can be transported to the central nervous system (CNS) via the vagus nerve or blood circulation (gut–brain axis), which affect the CNS function [3]. The above-mentioned monosaccharides, disaccharides, and the widely consumed rapidly digestive starch (RDS) after gelatinization could be promptly hydrolyzed in the small intestine, while slowly digestive starch (SDS) and resistant starch (RS) might delay digestion.

GCs can induce different glycemic responses. Elevated postprandial blood glucose level, a major risk factor for prediabetes and type II diabetes (T2DM), has emerged as a global epidemic [4]. The rapidly hydrolyzed carbohydrates can cause dramatic increases and decreases in blood glucose, resulting in severe glycemic fluctuations. Conversely, delaying starch digestion helps maintain glycemic homeostasis, thereby alleviating metabolic diseases [5,6]. Frequent and severe glycemic fluctuation is the embodiment of glycometabolism disorder, which is directly related to cognitive impairment [7,8]. Experiments on children have proven that sufficient glycemic supply induced by an acute increase in glycemic levels has a short-term (throughout the morning) positive impact on memory function [9,10], while a stable glycemic level can lead to better cognitive performance in the long term [11]. However, some of the crucial issues have not been fully elucidated yet. Most in vivo studies on the mechanism of glycemic fluctuations affecting cognitive function are based on typical metabolic diseases, such as obesity [12] and diabetes [8], while there is a lack of studies on the long-term impact of food-borne glycemic fluctuations on cognitive functions.

Therefore, comprehensive studies on the association between the reported GCs’ composition, gut microbiota, glycemic fluctuation patterns, glucose metabolisms, and neuron functions are highly imperative, which will help us to understand the relationship between carbohydrate consumption and cognitive functions and to conduct new studies from a more scientific perspective.

## 2. Glycemic Carbohydrate Digestion Rate and Glycemic Fluctuation Patterns

The monosaccharides can be absorbed directly into the small intestine, while disaccharides are converted into monosaccharides by α-glucosidases at the brush border of the small intestine mucosa. As for oligosaccharides, malto-oligosaccharides (MOS) have lower permeability than glucose, which can prolong the absorption time and energy supply, resulting in smoother glycemic fluctuations [13]. Isomaltooligosaccharides (IMO) can also be slowly hydrolyzed by α-glucosidases and induce a slight glycemic response [14]. Starches can be gradually hydrolyzed into glucose in the digestive tract by amylase and then absorbed into the blood, which is the main source of exogenous glucose causing glycemic fluctuations. According to previous studies, the digestive rates and glycemic index (GI) values of RDS, SDS, and RS were reduced sequentially, and the former induced significantly more dramatic glycemic fluctuations than the latter two [15,16,17]. α-Amylase first hydrolyzes starch structures to linear MOS and branched α-limit dextrins. A clear trend has been found toward incrementally slower digestion as α-limit dextrins become larger in size with more branched points [18]. As an incomplete hydrolysate of starch, maltodextrins can also be divided into rapidly, slow digestible, and resistant maltodextrins based on their anti-enzymatic ability, leading to different degrees of glycemic response [19].

Hyperglycemia, hypoglycemia, and glycemic fluctuation (the non-stable state in which blood glucose levels oscillate between peaks and valleys over a period of time) can be induced by inappropriate dietary habits and relevant diseases like obesity and diabetes. GI and glycemic load (GL = GI × amount of carbohydrate) are important indicators of the effect of food-induced glycemic response on CNS, and the effect of carbohydrates on blood glucose concentration. Researchers have found that high-GI carbohydrates are more likely to induce food addiction, suggesting that consumers may fall into a vicious circle of high-GI carbohydrates intake, addiction, and increased consumption [20]. However, no such relation between GI and digestion rate or glycemic response has been reported. Some SDS can retard digestion and provide a stable glycemic supply, maintaining energy metabolism homeostasis in vivo [21], but have high GI values due to their complex molecular structure and low contents of digestion-resisting components [22,23]. Extended blood glucose index is a potential index to evaluate the degree of slow and sustainable release of glucose from GCs [23]. Overall, unstable glycemic status can significantly increase the pressure to control glucose homeostasis, leading to inflammation, insulin resistance (IR), glucose, and other relevant metabolism disorders, resulting in neurocognitive impairment. It is worth noting that reducing carbohydrate intake is not an ideal way to reduce glycemic fluctuation or GL. Studies have shown that a reasonable intake of carbohydrates (45% to 60% of daily energy intake) is crucial to ensuring a balanced intake of macronutrients and maintaining metabolic health [24]. Some emerging special dietary patterns such as energy-restricted/intermittent fasting diets [25] and ketogenic diets [26] have reported positive significance in assisting intervention under specialized supervision. However, they may carry risks when there is a lack of supervision or for some special populations. It has been reported that both may lead to nutrient deficiencies over a prolonged period for patients with multiple sclerosis [27]. Intermittent fasting with unreasonable timing can damage immune functions and increase inflammatory responses [28]. For most individuals, a ketogenic diet may lead to decreased appetite and insufficient nutrient intake, which exacerbates the long-term health risk of maintaining this pattern [26,29]. Additionally, a return to a regular-carb diet after a prolonged period of low-carb diet is more likely to cause acute negative consequences like disease-related carbotoxicity [6]. Therefore, the research and development of smooth glycemic response carbohydrates could be a potential approach to reducing the potential hazards of glycemic carbohydrate consumption.

## 3. Susceptible Population for Diet-Induced Neurocognitive Impairments

### 3.1. Early and Late Life of Healthy Individuals

Dietary intervention is a moderate, time-consuming process. Some populations are vulnerable to glycemic carbohydrate-induced changes in the gut microenvironment or glycemic fluctuation-induced neurocognitive functions. The CNS is immature and susceptible to the influence of external factors in early life. Generally, the human brain maintains a high glucose utilization rate at between 4 and 10 years old due to the high metabolic rate and intense learning tasks, and then the rate gradually decreases and reaches an adult value between 16 and 18 [30]. Evidence suggests that high-sugar diet (HSD)-induced gut microbiota could impair hippocampus-dependent neurocognition in rats during childhood and adolescence [31,32,33]. Conversely, it was reported that HSD did not significantly affect cognitive functions in adult rats [34]. A follow-up study on dietary patterns found that a sugar-rich diet during early childhood can lead to smaller cerebral white matter volume, while a whole grains-rich diet can lead to larger cerebral gray matter volume and a larger surface area of the prefrontal cortex [35]. These results confirm that dietary patterns play a critical role in neurological health development and cognitive performance during adolescence and adulthood.

Studies have also reported a decline in age-related cerebral glucose metabolism and attenuated counter-regulatory responses in middle age [30]. Additionally, aging can exacerbate high-fat diet (HFD)-induced neuroinflammation and associated cognitive impairments [36]. Aging can lead to energy metabolism disorders, mitochondrial dysfunction, and the gradual failure of defense mechanisms against abnormal proteins. Hence, the neurons in the elderly are vulnerable to injury, leading to mild cognitive impairment, a concept of cognitive impairment intervening with normal aging and early dementia [37], or Alzheimer’s disease (AD) [38]. It is reported that excessive dietary sugar intake is significantly associated with AD risk in older women [39]. Aging can also induce hippocampal inflammatory responses due to a high-calorie diet and worsen cognitive phenotypes [36]. On the other hand, it is reported that the aging trajectory of the gut microbiota is associated with metabolic diseases and neurodegenerative diseases in a chronological age-dependent manner [40,41]. Evidence shows that, compared with young mice, aged mice show spatial memory deficits, accompanied by gut microbiota imbalance, increased intestinal permeability, and increased peripheral inflammatory cytokines [42]. The transplantation of fecal bacteria from aged rats to young rats can significantly damage the cognitive functions of the latter, which might be attributed to the increased levels of proinflammatory cytokines and oxidative stress (OS) related to the elderly gut microbiota, the changes in the hippocampal synaptic structure, and the decreased expression of brain-derived neurotrophic factors (BDNF) [43].

### 3.2. Patients with Metabolic Diseases

Neurons are susceptible to the effects of external glucose concentration due to their inability to store glycogen. Therefore, some metabolic diseases with blood glucose fluctuations as one of the main symptoms are highly susceptible to cognitive decline (Table 1). Diabetics are prone to developing diabetes-associated cognitive decline (DACD) [44]. Some typical symptoms of diabetes, such as impaired glucose tolerance, IR, chronic hyperglycemia, glycemic fluctuations, and the resulting deterioration in gut microbiota might contribute to cognitive impairments [45]. Additionally, as a metabolic disease induced by prolonged, inappropriate dietary habits and an important driving force of T2DM, obesity has been proven to be a potent risk factor for the onset and progression of several neurological disorders [46]. Studies have shown a negative correlation between obesity indicators and some cognitive indicators in the elderly, such as episodic memory, verbal learning, AD, and vascular dementia [47]. However, it is worth noting that these findings may be interfered with by different factors, such as the cause, degree, and stages of obesity development. Since obesity cannot be distinguished from complications, such as hyperglycemia and IR, studies on the correlation between obesity and cognitive decline are sometimes controversial. Typically, obesity could induce OS by disrupting the adipose microenvironment and mediating low-grade chronic inflammation and mitochondrial dysfunction, leading to cognitive decline [48]. It is reported that obesity-related cognitive decline is accompanied by a reduction in the dendritic spine density and synaptic sites associated with obesity-induced phagocytosis of synapses [49,50].

Patients with these metabolic diseases have reduced diversity of gut microbiota due to their inappropriate dietary habits and metabolic homeostasis disorders. Similarly, patients with diabetes or hyperglycemia often have reduced diversity of gut microbiota and increased gut barrier permeability [55,56]. Compared with healthy individuals, obese patients typically have higher *Firmicutes*, lower *Bacteroidetes*, and lower SCFA production in their gut, which have been proven to be potent risk factors for cognitive decline [57,58].

## 4. Microbiota Remodeling and Neurocognitive Functions

### 4.1. Glycemic Carbohydrate Diet-Induced Microbiota Remodeling

Gut microbes can respond rapidly to changes in dietary patterns. The remodeling effect of carbohydrates on gut microbes and microenvironment is significantly related to their digestive properties (see Appendix A). RS, SDS, and IMOs are the primary carbohydrates available to bacteria in the colon. The entry of RS-containing starch into the colon provides the preferred fuel for microorganisms and helps improve the gut microenvironment. Dietary RS supplementation can attenuate hyperglycemic, hyperinsulinemic, and hyperlipidemic responses by restricting gluconeogenesis, bolstering glycogenesis, and maintaining glucose and lipid homeostasis, satiety, and colonic health [59,60]. Different types of RS can lead to different dominant bacteria and SCFAs in the gut. Recent studies have confirmed that HSD also affects gut microbial composition [33].

### 4.2. Microbial Metabolites and Neurocognitive Functions

Accumulating studies have reported the effects of HSD, high-sugar and fat diets (also known as Western diets), and ketogenic diets on cognitive functions through the gut–brain axis. Currently, the negative impact of excessive intake of high-sugar beverages and desserts on cognitive functions in children and adolescents warrants further exploration. Studies have shown that gut microbiota optimization could help improve DACD or mild cognitive impairment [44] and AD [61]. It was reported that five different genera, namely *Bifidobacterium*, *Bacteroides*, *Faecalibacterium*, *Akkermansia,* and *Roseburia,* were negatively associated with T2DM, while *Ruminococcus*, *Fusobacterium*, and *Blautia* were positively associated with T2DM [62]. The abundance of these microorganisms was influenced by carbohydrate intake to some extent (Appendix A). The prolonged signal communications between the gut microbiota and CNS through microbial metabolites often interfere with host metabolism and cognitive functions. RS and SDS remodel the gut microbiota and increase the production of some beneficial metabolites, such as neurotransmitters, glycerophospholipids, and SCFAs, which enhance the integrity of both IEB and BBB. These metabolites can be transported to the CNS through blood circulation or the vagus nerve to help preserve healthy neurocognitive functions.

#### 4.2.1. SCFAs

Colonic microbiota can use complex carbohydrates such as cellobiose and starch for growth and proliferation and significantly increase the production of SCFAs, such as acetate, propionate, and butyrate. The delivery of optimal levels of SCFAs to the brain can confer neuroprotective effects through the promotion of cell proliferation, neuroblast differentiation, and BDNF expression, as well as by reducing systemic inflammation. For example, sodium butyrate (300 mg/kg/d, injected for 3 weeks) was reported to be used as an effective memory-enhancing drug for mice [63,64].

Additionally, SCFAs help maintain the integrity of the intestinal epithelial barrier (IEB), thus preventing the entry of toxic microbial metabolites from bacterial-derived LPS into the body’s circulation [65]. It is reported that the gut microbiota has significant effects on gut permeability [66] and blood brain barrier (BBB) integrity [67]. Impairments to both barriers increase the chance of adverse substances entering the body’s circulation and the CNS. This promotes metabolic disturbances and exacerbates systemic inflammation, disrupting the neurocognitive function. Gut function impairment, such as a diminished diversity of gut microbiota and heightened permeability of the gut barrier, is more prevalent in the obese, diabetic, hyperglycemia, and elderly populations [68,69]. Akkermansia muciniphila colonizes on the mucus layer of the gastrointestinal tract. It can degrade mucin and produce SCFAs, such as acetate and propionate, to increase mucus production and intestinal epithelial cell regeneration, maintaining the dynamic stability of the mucus layer and improving IEB function [70,71]. The abundance of this bacterium is lower in patients with obesity and T2DM [72,73], while the intake of RS can increase its abundance [74]. SCFAs, represented by butyrate produced by RS fermentation in the gut, have been reported to pose a protective effect on barrier integrity [33,71,75]. Conversely, the reduced production of SCFAs induced by high fructose feeding leads to damaged IEB and elevated serum endotoxin levels, resulting in hippocampal neuroinflammatory responses and further neuronal loss [76].

#### 4.2.2. Neurotransmitters

Certain gut bacteria can influence the expression of BDNF and neurotransmitters (such as glutamate, GABA, 5-HT, dopamine, and acetylcholine) and associated host behavior in a vagus-dependent manner or via blood circulation by influencing the synthesis of neurotransmitters in the gut cells [77,78]. For instance, the gastrointestinal production of 5-hydroxytryptamine (5-HT) [79] is directly related to CNS functions like neurocognition and depression. Intestinal *Alistipes* levels are related to the carbohydrate digestive properties, with sugar and RDS intake increasing their abundance and SDS or RS intake decreasing their abundance (Appendix A). *Alistipes* can hydrolyze tryptophan, a precursor of serotonin, to indole, decreasing the availability of 5-HT [80], while the intervention of *Bifidobacterium* spp. or *Clostridium* spp. increases the availability of 5-HT and its precursor levels [79,81].

#### 4.2.3. Glycerophospholipids

Neural membranes contain several classes of glycerophospholipids essential for membrane integrity and stability [82,83]. A study on bumblebees shows that the production of glycerophospholipids associated with the intestinal bacterial phosphoenolpyruvate-dependent phosphotransferase system could significantly improve long-term memory capacity [84]. These glycerophospholipids are synthesized by glycolipid metabolism in vivo or in the gut bacteria and are subsequently absorbed into the body. Glycerophospholipids can generate second messengers through further degradation, such as arachidonic acid, platelet activating factor, and diacylglycerol, which are strongly associated with synaptic plasticity and cognitive ability [83].

## 5. Glycemic Fluctuations and Neurocognitive Functions

The brain only accounts for 2% of the body’s weight but consumes 20% of the oxygen and 25% of the glucose of the whole body. Energy metabolism disorders of neurons that exist in AD appeared earlier than β-amyloid (A-β) deposition [85]. As the primary peripheral organ of glucose metabolism, the liver undergoes similar changes to the CNS under unstable glycemic levels. Glucose fluctuation can affect neuronal functions by disturbing glucose metabolism, inducing neurotoxic substances, or interfering directly with calcium homeostasis.

### 5.1. Glycemic Fluctuation-Induced Glucose Metabolic Disorders

Glucose metabolism is the core of connections among glycemic levels, other metabolic pathways, and neurocognitive functions. In the CNS, a close energy and substance coupling relationship exists between astrocytes and neurons. As for energy supply, astrocytes rely mainly on glycolysis, and neurons use oxidative phosphorylation and the tricarboxylic acid (TCA) cycle [86]. Neurons are more sensitive to the perception of glucose fluctuations than astrocytes due to their high energy demand, higher affinity, and the higher turnover rate of glucose transporter-3 (GLUT3) for glucose than GLUT1 [87]. As shown in Figure 1A, when astrocytes are exposed to high glucose, the increased pyruvate might be converted to lactate. It can be shuttled into neurons to be converted into pyruvate and then into the TCA cycle [88]. Glutamate is secreted during neuronal excitation induced by elevated glucose levels, and then absorbed by astrocytes. This process activates Na^+^/K^+^ATPase, a key factor in promoting glucose uptake by GLUTs. In vitro studies suggest that astrocytes can detect the synaptic activity of glutamate neurons through glutamate transport activity and convert it into metabolic signals, so that the nerve cells can take up glucose as needed [89,90].

Dysregulation of glucose metabolism can lead to pathogenesis and T2DM complications, leading to IR and glycemic variability [91]. Long-term chronic hyperglycemia (>15 mmol/L) might lead to metabolic disorders and increased anaerobic metabolism [53]. Long-term stable or intermittent hyperglycemia can promote the generation of several neurotoxic substances. Correspondingly, recurrent non-severe hypoglycemia can lead to cognitive dysfunction in diabetics. Improper coordination of hypoglycemic drugs, diet, and exercise can easily lead to hypoglycemia (<3 mmol/L), an independent risk factor for DACD [92,93]. Hypoglycemia leads to insufficient energy supply, abnormal glucose metabolism, and reduced synaptic plasticity in the hippocampus [94]. Additionally, studies have shown that intermittent high glucose can lead to more severe chronic diabetes complications than sustained high glucose [95,96,97]. It has been reported that the repetition of hyper- and hypoglycemic cycles appears to contribute to IR, T2DM, and obesity [98]. In vitro studies have found that, compared with the cells cultured with constant high glucose, the cells cultured with fluctuating glucose are more prone to mitochondrial dysfunction and cellular DNA damage [99].

Although both GLUT1 and GLUT3 are non-insulin dependent, GLUT4 is highly expressed in the hippocampus in an insulin-dependent manner [100]. Disordered glucose metabolism may lead to IR and further cognitive decline. Then, the insulin-like growth factors-1 (IGF-1) can also bind and activate insulin receptors and subsequent pathways, which play an important role in brain neurogenesis, A-β clearance, and neuronal trophic support. Excessive insulin caused by IR competes with A-β for insulin-degrading enzymes, thereby reducing the degradation and consequent deposition of A-β [101]. Moreover, glycogen synthase kinase-3β (GSK-3β), one of the major tau kinases in the insulin-P13K-AKT signaling pathway, can be activated by IR [102,103]. IGF-1 resistance and enhanced GSK-3β activity can cause tau hyperphosphorylation, leading to synaptic dysfunction and neuronal apoptosis [104]. In addition, IR is also associated with OS and neuroinflammation.

### 5.2. Neurotoxic Substances and Neurocognitive Functions

Intracellular glucose fluctuation may activate several glycolysis-related pathways, such as the polyol, fructose metabolism, advanced glycation end products (AGEs), protein kinase C (PKC), and hexosamine pathways. Disordered glycolysis and some glycolytic bypass pathways can produce harmful glycometabolites and many common neurotoxic substances, such as AGEs, reactive oxygen species (ROS), and proinflammatory cytokines, thus affecting neurocognitive functions. In addition to causing direct damage to neurons, neurotoxic substances can increase the pathologic permeability of BBB by damaging the brain microvasculature and destroying the tight junction between the brain microvascular endothelial cells. As such, more toxic substances could enter the brain and induce more severe inflammatory responses and cognitive decline [105,106].

#### 5.2.1. Glycometabolites

Excess lactic acid and other acidic metabolites will be released by enhanced anaerobic metabolism in the astrocytes under chronic hyperglycemia, which might lead to acidosis and hypoxia damage to neurons and ultimately endanger the CNS [107,108]. In the polyol pathway, the first enzyme, aldose reductase (AR), has a lower affinity for glucose than HK in glycolysis. As shown in Figure 1B, the saturation of hexokinase (HK) under high glucose concentrations can activate the polyol pathway, producing sorbitol and fructose. The activated polyol pathway induces intracellular accumulation of sorbitol, resulting in tissue swelling and direct tissue toxicity due to the poor membrane permeability of sorbitol [100].

Furthermore, excessive intake of fructose and endogenous fructose production can over-activate the fructose metabolic pathway, leading to various metabolic disorders and even dependence on sugar consumption [109,110,111]. The intake of GCs, such as glucose and maltodextrin [112], can also stimulate this process. Evidence suggests that excessive activation of the cerebral fructose metabolism is associated with cognition decline [113]. The activation of ketohexokinase (KHK) in this pathway leads to a significant consumption of intrahepatic phosphate and ATP levels, and an increase in uric acid levels [113,114]. Uric acid in the serum can cross BBB into the brain and induce neuronal death by stimulating mitochondrial OS and neuroinflammation, thereby resulting in cognitive decline [115,116]. Additionally, fructose promotes the production of AGEs, as discussed later.

#### 5.2.2. AGEs

Intracellular AGEs are harmful to cells, as they induce the loss or alteration of normal cellular protein or lipid functions. Increased AGEs promote axonal atrophy and/or degeneration and reduce the innate ability of neurons to self-repair [117]. Additionally, it increases matrix metalloproteinase production, exacerbating nerve fiber damage further [118]. High glycemic levels may promote the formation of AGEs in the following ways (Figure 2).

Firstly, hyperglycemia results in non-enzymatic glycation of excess glucose with amino acids, proteins, lipids, or nucleic acids. The Maillard reaction between the carbonyl group and amino group generates a Schiff base and then obtains the relatively stable ketoamine compounds, namely Amadori products, through Amadori rearrangement. Glucose-derived AGEs (Glu-AGEs) can be formed after further rearrangements [119]. Secondly, excess glucose activates the polyol pathway and consumes NAD^+^ during the oxidation of sorbitol to fructose, resulting in an increase in the ratio of NADH/NAD^+^. NAD^+^ is the coenzyme of glyceraldehyde-3-phosphate dehydrogenase (GAPDH) in glycolysis. Hence, excessive consumption of NAD^+^ might induce the accumulation of its upstream substrates represented by glyceraldehyde-3-phosphate (G3P). G3P can be converted into highly reactive methylglyoxal, a precursor of AGEs [100,120]. Similarly, fructose produced by the polyol pathway can also form Fru-AGEs. In addition, glucose-induced OS can activate an NAD^+^-dependent DNA repair enzyme called poly(ADP-Ribose) polymerase (PARP) by inducing DNA damage, which inhibits the GAPDH activity by catabolizing the NAD^+^-derived ADPR [87,121]. Thirdly, the produced fructose can be phosphorylated to fructose-3-phosphate (F3P) and converted into 3-deoxyglucosone further. Both compounds are effective glycation agents contributing to AGEs formation [120,122]. However, studies on F3P promoting AGE formation are only processed for the rat lens. There is no evidence of the occurrence of this process in hippocampal neurons.

#### 5.2.3. ROS

The brain expends a significant amount of oxygen to sustain its normal physiological functions, yet it possesses higher lipid levels and lower concentrations of antioxidant enzymes compared to other organs [123,124]. Consequently, it is highly susceptible to oxidative stress (OS), a pivotal factor contributing to neuronal degeneration. The high sensitivity of neurons and glial cells to OS is mainly due to the excessive production of ROS in the mitochondrial respiratory chain under high glucose concentrations leading to OS, neuroinflammation, and apoptosis, as summarized in Figure 2.

In the early stages of hyperglycemia, elevated glucose level provides more fuel for the TCA cycle, thereby increasing the production of electron donors (NADH). The resulting electron transfer chain overload might lead to electron leakage and ROS formation in the mitochondria. In the cytoplasm, excessive glucose may activate the polyol pathway, depleting NADPH, a cofactor of glutathione reductase, leading to OS [87]. Then the PKC pathway can be activated by an increased synthesis of diacylglycerol, a PKC agonist [120]. PKC is known to induce cellular ROS by activating NADPH oxidases [125]. Additionally, the interaction between extracellular AGEs and their receptor RAGE can activate diverse signal transduction cascades and induce the production of downstream ROS [126]. The non-enzymatic glycation of proteins is accompanied by an oxidation reaction. Under aerobic conditions, Amadori products can generate products such as pentosidine through the oxidation pathway and release free radicals to cause OS [127]. Intracellular AGEs may aggravate OS, which itself up-regulates RAGE through NF-κB activation, thereby creating a vicious cycle [87]. Moreover, 3-phosphoglycerate generated by glycolysis can be gradually metabolized to L-serine through the serine synthesis pathway (SSP) with glutamate consumption and α-ketoglutarate production [128,129]. Serine is a synthetic precursor to glycine, which can be metabolized to produce the antioxidant glutathione to protect neurons from OS. However, higher levels of glutamate and lower levels of glycine have been reported in the plasma of diabetics, contributing to higher OS levels [130,131]. Particularly, it has been reported that the enhanced impairment of intermittent high/low glucose can activate OS through the endoplasmic reticulum (ER) stress response than stable high/low glucose [34,96,132].

#### 5.2.4. Proinflammatory Cytokines

Persistent, chronic, and low-grade inflammation is one of the common symptoms of most neurodegenerative diseases. As shown in Figure 2, constant or oscillating high glucose may up-regulate the expression of proinflammatory cytokines and induce the inflammatory cascade reaction, leading to apoptosis of the nerve cells. In addition, it is reported that elevated levels of inflammatory factors like tumor necrosis factor-α (TNF-α) in T2DM mice lead to reduced expression of phosphoserine aminotransferase1 in SSP, inhibiting the synthesis of serine and tribbles homolog 3, thus resulting in decreased insulin sensitivity, or even IR [131]. 

AGEs and their binding with RAGE can induce an inflammatory cascade by activating the nuclear factor kappa B (NF-κB) pathway [126,133]. ROS can also activate the NF-𝜅B pathway and promote the production of inflammatory cytokines. The binding of inflammatory cytokines like TNF-α and their receptors can also trigger ROS formation through various potential pathways [134]. PKC can activate the NF-𝜅B pathway by activating the mitogen-activated protein kinase (MAPK) pathway [126]. Then, high glucose can inhibit GADPH and induce the accumulation of upstream products such as fructose-6-phosphate (F6P), which can activate another bypass of glycolysis called the hexosamine pathway. F6P can be converted into glucosamine-6-phosphate and subsequent uridine diphosphate-*N*-acetylhexosamine (UDP-GlcNAc) under glutamine F6P aminotransferase. UDP-GlcNAc can combine with serine and threonine residues of the transcription factor Sp1, resulting in an increased O-GlcNAcylation of SP1 and increased expression of Sp1-dependent genes, such as the transforming growth factor-β, which may result in neurovascular impairment [100,121]. In addition, the peroxisome proliferator-activated receptors (PPARs) play an important role in regulating energy metabolism. Several studies on peripheral tissue cells have reported that hyperglycemia down-regulates the expression of PPARs and induces inflammatory responses [135,136]. In the gut, the intake of RS has been shown to inhibit intestinal inflammation by reducing the production of pro-inflammatory mediators (such as TNF-α and NF-κB) and SCFAs and increasing the expression of the nuclear transcription factor PPARγ [137]. Additionally, research on peripheral tissues has reported that glucose fluctuations can increase the expression of inflammatory factors than hyperglycemia, providing insights into the mechanisms of the effects of glucose fluctuations on neuronal cells [138].

### 5.3. Calcium Overload and Neurocognitive Functions

Glucose level is not only an energy signal but also a status signal. Calcium is a common intracellular second messenger. With increased glucose acting as an upstream extracellular signal binding to its receptor, Ca^2+^ enters the cell and triggers a significant increase in the cytoplasmic Ca^2+^ concentration [139,140]. Severe glucose fluctuations in the circulation cause persistent hyperglycemic stimulation in the nerve cells, leading to the dysregulation of intracellular calcium homeostasis and neuropathy by affecting the expression and function of Ca^2+^ channels (for Ca^2+^ influx) or pumps (for Ca^2+^ efflux) [140,141]. An in vitro study demonstrated that a high extracellular glucose level (50 mM) significantly increased the cytoplasmic Ca^2+^ concentration and gene expression of the store-operated calcium channel (SOCE)-related protein STIM1 in rat primary hippocampal neurons, mediating more Ca^2+^ influx through the calcium release-activated calcium channel protein Orai on the plasma membrane [140,142]. In addition, the activation of PKC can promote Ca^2+^ influx by activating the voltage-gated calcium channels (VGCCs) on the plasma membrane further [143]. Disruption of Ca^2+^ homeostasis is mainly manifested by a large Ca^2+^ influx due to altered cell membrane permeability. The ER is the main storage site for Ca^2+^ in eukaryotic cells. Ca^2+^ influx triggers Ca^2+^ release from the ER, leading to intracellular Ca^2+^ overload, which can induce neuronal hypofunction or structural damage in the following ways (Figure 3).

#### 5.3.1. Disrupt Energy Metabolisms Associated with Ca^2+^

Mitochondrial Ca^2+^ overload, triggered by cytoplasmic Ca^2+^ overload, allows the formation and deposition of calcium phosphate in the mitochondria, thereby affecting the oxidative phosphorylation capacity and leading to reduced ATP synthesis and impaired energy production [144,145]. Normally, Ca^2+^ flowing into the cytoplasm can be partially bound to calmodulin (CaM) and activate the target enzymes such as adenosine 5‘-monophosphate (AMP)-activated protein kinase (AMPK) through prior activation of CaMKII (Ca^2+^/CaM-dependent protein kinase). Impaired mitochondrial ATP production and CaMKII overactivation due to Ca^2+^ overload can over activate AMPK, adversely affecting the regulation of energy balance, particularly in the brain [139]. 

#### 5.3.2. Trigger Neuron Injury and Apoptosis

Above all, increased ER Ca^2+^ depletion due to Ca^2+^ overload can lead to ER stress. It is reported that ER stress-mediated apoptosis is involved in hyperglycemia-induced synaptic and neuronal injury of the hippocampus [146]. Continuous ER stress can activate caspase-12 and lead to apoptosis by increasing the expression of calpain, a calcium-dependent protease [147]. A higher glucose level can also increase the ratio of Bax/Bcl-2 on the membrane and activate caspase 3 [140]. Furthermore, Ca^2+^ overload leads to excessive activation of intracellular Ca^2+^-dependent phospholipases, calpain, and nucleic acid endonucleases in the nucleus, thus destabilizing the cell membrane and cytoskeleton and causing DNA damage [145]. In addition, the increased pro-apoptotic protein Bax caused by high glucose can translocate to the outer membrane of mitochondria [148]. The loss of the mitochondrial membrane potential (ΔΨm ↓) and the leakage of electrons/mitochondrial Ca^2+^ overload-induced electron donors can lead to the formation of ROS. The increased Bax, the collapse of ΔΨm, and the resulting increase in ROS can induce the formation of the mitochondrial permeability transition pore, which can increase the release of cytochrome C from the mitochondria into the cytosol, thereby activating caspase 3 and initiating the mitochondria-dependent intrinsic apoptotic pathways [149,150,151].

#### 5.3.3. Promote Neurotoxic Protein Accumulation

Ca^2+^ overload can increase the neuroinflammation levels and A-β deposition by activating the calcium homeostasis modulator proteins [152]. A-β can also lead to mitochondrial Ca^2+^ dyshomeostasis and mitochondrial dysfunction [148]. Excessive cytosolic Ca^2+^ can induce tau hyperphosphorylation, while tau contributes to mitochondrial Ca^2+^ overload in neurons [151]. The intracellular accumulation of phosphorylated tau can trigger nuclear Ca^2+^/CaMKIV signaling, aggravating tau hyperphosphorylation and promoting neurodegeneration [153].

### 5.4. Glucose Fluctuations Trigger More Severe Impairments

Accumulated studies have confirmed some credible molecular mechanisms of cellular damage induced by glycemic fluctuations, such as triggered inflammatory responses and OS. However, the reason why glucose fluctuations can trigger more severe impairment than constant high/low glucose is still unclear. It is inferred that an organism possesses a certain degree of environmental adaptability and develops homeostasis adapted to the environment in which it is located. Persistent high/low blood glucose is a relatively unhealthy but stable environment. In this new environment, the organism tends to counteract the toxic effects of glucose through many feedback regulatory mechanisms to reduce impairment. However, glycemic fluctuation is a variable or even irregular process, making it difficult to develop adaptive stabilization mechanisms and feedback regulation, leading to disrupted homeostasis and accumulated impairments.

### 5.5. The Interaction between Glycemic Fluctuations and Gut Microbiota

Hyperglycemia or severe glycemic fluctuations can also affect the gut microenvironment, thereby promoting the adverse effects of glycemic fluctuations and causing more serious impairment. However, this phenomenon has not received much attention. Studies have confirmed that diabetes and hyperglycemia, which are characterized by glucose variability, negatively affect the gut functions, as generally manifested by reduced gut microbial diversity, weak gut barrier integrity, and increased pathogenic bacterial diversity [55,56,58]. Brittle diabetes (BDM) is a kind of unstable diabetes characterized by severe glycemic fluctuations. As such, the difference between BDM and non-Brittle T2DM can reflect the influence of glycemic fluctuation as an independent factor on gut microbiota and cognitive ability. BDM has been associated with a lower abundance of *Akkermansia muciniphila*, *Fusobacterium*, and *Prevotella copri*, with a corresponding enrichment of *Bacteroides vulgatus* and *Veillonella denticariosi* [154,155]. As discussed earlier, a significant decrease in the abundance of *A. muciniphila* induced by glycemic fluctuations could disrupt the intestinal barrier integrity. Additionally, hyperglycemia can disrupt the integrity of tight and adherent junctions through overexpression of GLUT2 [156]. The increased intestinal barrier permeability and GLUT2 overexpression lead to the formation of a vicious cycle, which can accelerate the absorption rate of carbohydrate digestion products, thereby interfering with glycemic fluctuations and contributing to cognitive impairment.

## 6. Carbohydrate Dietary Strategies Beneficial for Neurocognitive Functions

Table 2 summarizes the studies for different types of GC intervention-induced changes in neurocognitive functions. To facilitate a balanced, prolonged nutritional intake and reduce the health risks associated with specific dietary patterns, it is recommended to ensure an appropriate proportion of carbohydrate daily intake [24], especially for populations with a tradition of high carbohydrate consumption such as Asians [157]. Given that cognitive decline frequently coincides with certain metabolic disorders, there has been a significant focus on exploring optimal dietary patterns as a gentle and sustained approach to either enhance cognitive development or prevent cognitive decline. SDS, RS, and many whole grains could reduce the risks of these metabolic diseases through a smooth postprandial glucose response and positive regulation of the gut microenvironment, thus hindering and delaying the related cognitive decline [158,159].

### 6.1. RS Selection

The RS classification method provides references for selecting and processing starch-based foods. As a fermentable dietary fiber, RS provides effective assistance in shaping a healthier gut environment and a stable blood glucose response [174]. RSI suggests that non-refined carbohydrates that remain intact in the cell wall structures, such as whole grains, often have higher dietary fiber content and lower GI values. RSII presents naturally in some plants, such as raw potatoes, green bananas, and high-amylose corn. RSIII suggests that the way food is processed and cooked is also important for RS retention. For example, starch-based foods cooled after cooking have a higher RS content due to the retrogradation of starch; baked potatoes have higher RS than boiled potatoes due to less moisture content during heat treatment [175]. Zhanggui Wang et al. reviewed the methods of yielding RSIII from different sources [176]. In food processing, the preparation of RSIV&V requires some physical and chemical modifications, usually added as additives to improve the functional characteristics of the products.

### 6.2. SDS Selection

Slow-digesting maltodextrin and starch have emerged as the current research hotspots. Compared to RS, SDS can maintain a stable postprandial glucose response while providing a sufficient glucose supply. It is reported that A-type raw cereal starches are slowly digested with >50% SDS by the Englyst test [177]. Currently, whole grains and legumes are recognized as the ideal dietary options with distinct health benefits, including increased cognitive functions. A study followed the dietary habits of 516 young adults (32.03 ± 5.96 y) and assessed their cognitive performance at midlife (49.03 ± 4.86 y) [178]. The findings revealed a direct correlation between increased consumption of whole grains in the diet and enhanced cognitive performance. Another investigation found that a lower dietary intake of whole grains was associated with higher inflammatory markers and decreased cognitive ability in the elderly [179]. Generally, the cognitive protective effects of whole grains are associated with smooth glycemic responses due to their slow digestive properties, beneficial gut functions, and higher abundance of functional phytochemicals.

Dietary fiber is positively associated with cognitive performance among prepubertal children [180]. The abundance of dietary fiber in whole grains and the encapsulation effect of the cell wall on starch particles are the key factors in retarding the carbohydrate digestion rate, thereby maintaining postprandial blood glucose stability and improving the gut microenvironment [181]. The abundance of β-glucan in whole grain oats and barley has been shown to prevent cognitive disorders, which may be related to its involvement in maintaining the intestinal barrier function, countering HFD-induced microglia activation and neuroinflammation, and regulating the gut microbiota [162,182]. Additionally, β-glucan can increase the viscosity of the chyme, limiting enzyme accessibility to carbohydrates, reducing the glucose absorption rate through the intestinal wall, and limiting the post-prandial glycemic response [159]. Buckwheat whole flour (BWF) intervention over the long term has been shown to successfully mitigate age-related cognitive decline in mice [163]. In a previous study, exposure to BWF significantly increased the expression of postsynaptic Arc and postsynaptic density protein-95 (PSD-95) and the mature neuronal marker NeuN in the hippocampus due to the increased abundance of *Lactococcus* and *Ruminiclostridium* in the gut. Additionally, the beneficial effect of whole grains on cognitive ability is largely related to their functional phytochemicals. The 5-heptadecylresorcinol in whole grain rye has been proven to improve cognitive functions and neuroinflammation in APP/PS1 transgenic mice [183]. Avenanthramides and avenacosides in whole grain oats have good anti-inflammatory and antioxidant properties [184]. The abundance of lipopolysaccharides in brown rice has been proven to improve cognitive functions in mice and elderly humans [161,185]. Ferulic acid in whole grains could achieve neuroprotective effects through anti-inflammatory, antioxidant, and glucose homeostasis regulating abilities [186].

It is important to highlight that individuals with limited chewing capacity and digestive functions, such as children and the elderly, need to be particularly mindful of the potential harm from the coarseness and hardness of whole grains and resistant starch (RS). Employing suitable pre-treatment methods for brown rice, such as hydrostatic treatment, is advisable to ensure its tenderness and ease of consumption [161].

### 6.3. Sweetener Selection

With the rapid development of society, people have gradually realized the adverse effects of excessive sugar or rapidly digested carbohydrate intake on health, including cognitive functions (Table 2). Studies have confirmed that excessive sugar intake, including glucose, sucrose, and fructose, during childhood can lead to cognitive decline compared to ungelatinized ordinary corn starch, an SDS with carbohydrates in standard chow (the study models involved are mice and rats, as shown in Table 2). Therefore, the current consensus is that reducing sugar intake can lead to better cognitive abilities under sufficient energy supply.

The invention and application of non-nutritive artificial sweeteners (NNSs) is an effective approach to reduce these risks. NNS is considered a healthier sweetener alternative due to its high sweetness, low-calorie content, and almost no involvement in body metabolism. Evidence suggests that NNSs can protect cognitive functions better than traditional sugars (Table 2). Nevertheless, there is still controversy over whether NNSs have adverse effects on cognition compared to water (control), which also generates no energy. For example, studies have found that NNS intervention leads to better cognitive abilities compared to sugar but had no significant difference compared to the water-fed group [164,173]. However, in another study, compared to the control, NNS (acesulfame, saccharin, stevia) intervention in adolescent rats induced hippocampal-dependent cognitive decline [172]. Another intervention for diabetic rats also showed that, compared with the control, the cognitive ability of rats taking aspartame solution was significantly reduced [171]. Recent reports discuss the potential negative effects of NNSs on appetite, metabolic health, and cognitive functions through various peripheral and central mechanisms [187,188,189,190]. Nevertheless, relevant evidence and conclusions are still controversial. Therefore, NNS, a traditional sugar substitute with a wide variety and insufficient evidence of long-term health effects, deserves further research.

### 6.4. Research Models on Diet-Induced Changes in Neuro Functions

Some research models can be drawn upon on how dietary patterns affect cognitive functions (Figure 4). Firstly, we propose studying the diet-induced changes in the gut microbiota and the correlation analysis with behavior, which could verify if these changes are the key intermediate factors between these two. The effect of highly correlated species on CNS can be further verified by subsequent animal experiments such as oral gavage of the target strain or fecal transplantation. The effect mechanisms mainly focus on the key metabolites and their signal pathways. The study approach generally involves metabolomic or transcriptomic analysis of fecal/intestinal contents/liver (place of origin), serum (transmission pathway), and target CNS regions such as the hippocampus (destination). Untargeted metabolomics allows broad-spectrum screening of differential metabolites between groups, complemented by the KEGG enrichment analysis to obtain the major metabolites affecting cognition. Targeted metabolomics is used for studies with clear targets or metabolic pathways. Forward and reverse validation based on target metabolites can increase the credibility of the study. Forward validation refers to feeding/gavaging/injecting the target metabolite to the laboratory animals based on the normal diet. Reverse validation can be performed by knocking out or silencing of genes for key enzymes or receptors for the metabolites and comparing the cognitive differences with control. Selection of study models is also important. Contrasting with the rapid and pronounced therapeutic effects typically expected from medical treatments, dietary interventions are generally appropriate for most healthy individuals or as a supplement to medical treatment in patients with specific diseases, marked by gentler effects and extended duration. Therefore, the effectiveness is generally demonstrated in populations with weaker or vulnerable cognitive functions, such as children and adolescents with immature brains in early life, diabetics, and the elderly prone to cognitive impairment.

## 7. Conclusions

Glycemic carbohydrate is a major and economical energy source for humans, but its inappropriate consumption can lead to various metabolic and neurodegenerative diseases, including cognitive impairment. In modern society, the rapid increase in mental labor and stress and the pursuit of immediate happiness feedback have led to a sharp increase in the demand for rapidly digestible carbohydrates, such as sugar and pregelatinized starch-based foods [191,192,193]. However, there is substantial evidence to suggest that GCs with different digestive characteristics can lead to differences in gut microbiota diversity and glucose fluctuation patterns, which have been demonstrated have an association with neurological impairment (Figure 5), especially for those susceptible populations, including those in early and late life and those suffering from metabolic diseases such as obesity and diabetes. Unhealthy glycemic fluctuations can negatively impact brain functions in various ways by disrupting the gut microbiota structure, causing metabolic disturbances and calcium overload, and producing neurotoxic substances. Therefore, it is imperative to adjust the dietary strategies to promote the development of neurocognitive functions better, including increasing the intake of RS and SDS as appropriate, choosing whole grains as a staple instead of refined carbohydrates, and reducing sugar intake while ensuring sufficient energy intake. Additionally, as discussed in this review, there is a lack of research on how long-term intake of starch with different digestive properties affects neurocognitive function. Moreover, the existing targeted research and mechanism studies remain inadequate, with some controversial conclusions arising from the wide variety of GCs. As such, achieving a more judicious selection of carbohydrates or mitigating health risks through scientific processing while considering sensory enjoyment is still worth further study.

## Figures and Tables

**Figure 1 nutrients-15-05080-f001:**
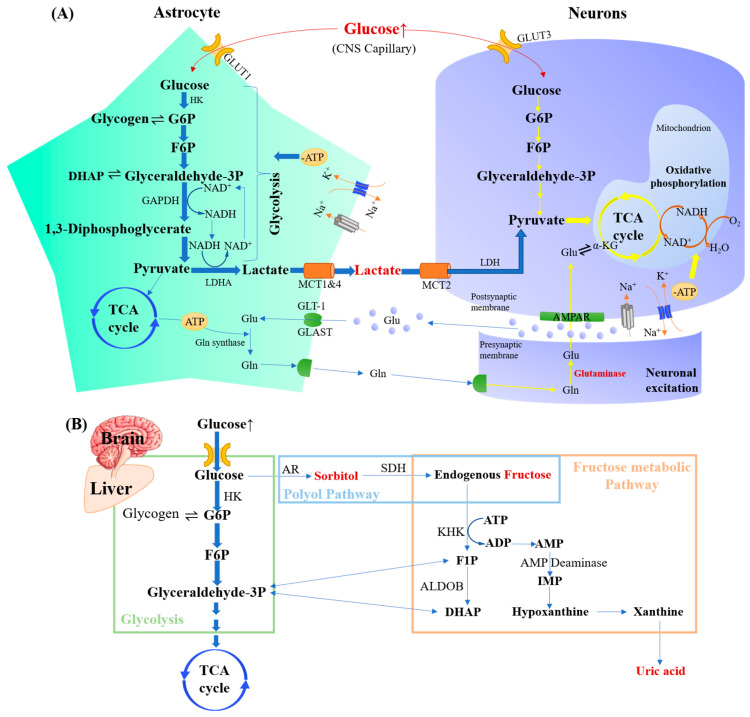
Glucose metabolism disorders and the production of harmful glycometabolites. (**A**) Glucose metabolism and the accumulation of lactate between astrocytes and neurons upon high glucose. Excessive glucose intake effectively activates glycolysis in astrocytes and TCA cycling and oxidative phosphorylation in neurons. Increased pyruvate produced in astrocytes might be converted to lactate, which can be then shuttled into neurons and converted into pyruvate. (**B**) The accumulation of sorbitol, fructose, and uric acid upon high glucose. When HK is saturated upon high glucose, the polyol pathway and fructose metabolism pathway can be activated so that the sorbitol, fructose, and the uric acid can be produced and accumulated. ADP, adenosine diphosphate; ALDOB, aldolase B; AMP, adenosine monophosphate; AMPAR, alpha-amino-3-hydroxy-5-methyl-4-isoxazole propionic acid receptor; ATP, adenosine triphosphate; DHAP, dihydroxyacetone phosphate; F1P, fructose-1-phosphate; G6P, glucose-6-phosphate; Gln, glutamine; Glu, glutamic acid; GLAST, glutamate aspartate transporter; GLT-1, glutamate transporter-1; GLUT, glucose transporter; IMP, inosine monophosphate; LDH, lactate dehydrogenase; LDHA, lactate dehydrogenase-A: MCT, monocarboxylate transporter; NAD^+^/NADH, nicotinamide adenine dinucleotide (oxidized/reduced); PSAT1, phosphoserine aminotransferase 1; SDH, sorbitol dehydrogenase.

**Figure 2 nutrients-15-05080-f002:**
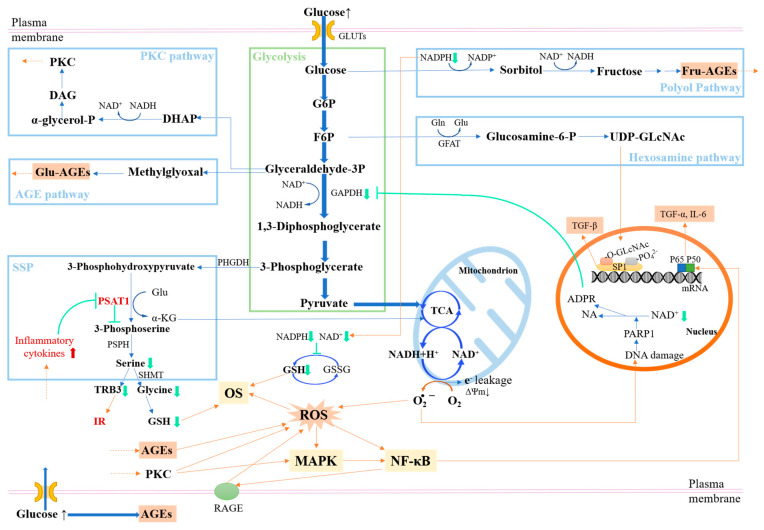
Neurotoxic substances triggered by a high glucose level. Fluctuating intracellular glucose may activate some glycolysis bypasses such as the polyol pathway, SSP, fructose metabolism pathway, AGE pathway, PKC pathway, and hexosamine pathway. They can produce a number of common neurotoxic substances, such as AGEs, ROS, and proinflammatory cytokines, thus affecting neurocognitive functions. α-KG, α-ketoglutarate; ADPR, adenosine diphosphate ribose; DAG, diacylglycerol; GFAT, fructose-6-phosphate amido transferase; GSH, antioxidant glutathione; GSSG, glutathione (oxidized); IL-6, interleukin 6; MAPK, mitogen-activated protein kinases; NA, nicotinic acid; NADP^+^, nicotinamide adenine dinucleotide phosphate (oxidized); NADPH, nicotinamide adenine dinucleotide phosphate (reduced); NF-κb, kappa-light-chain enhancer of activated B cells; O-GLcNAc, O-linked-N-acetylglucosaminylation; PSAT1, phosphoserine aminotransferase1; TGF, the transforming growth factor; TRB3, tribbles homolog 3.

**Figure 3 nutrients-15-05080-f003:**
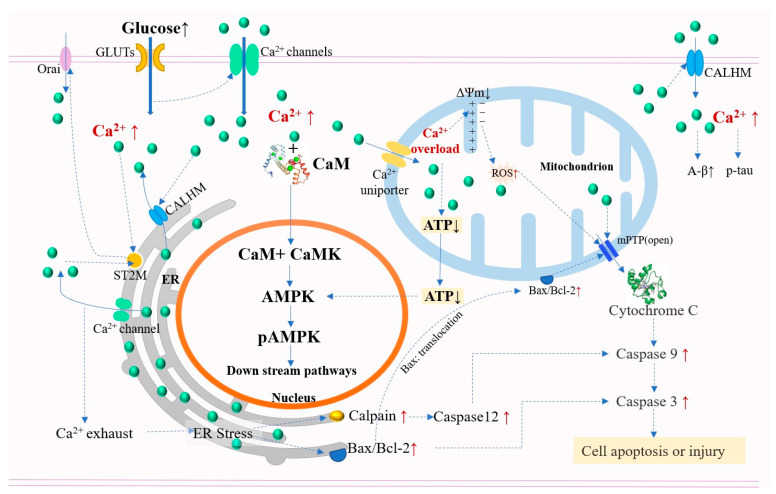
Calcium overload and its adverse effects on neurological function induced by high glucose stimulation. The increased extracellular glucose triggers a significant increase in the cytoplasmic Ca^2+^ concentration. Calcium overload may lead to disrupted energy metabolism, neuron injury/apoptosis, and neurotoxic protein accumulation. (AMPK, adenosine 5‘-monophosphate (AMP)-activated protein kinase; Bcl-2, B-cell lymphoma-2; Bax, Bcl-2-associated x; CALHM, calcium homeostasis modulator protein; mPTP, mitochondrial permeability transition pore; Orai, calcium release-activated calcium channel protein; ST2M, stromal interaction molecule).

**Figure 4 nutrients-15-05080-f004:**
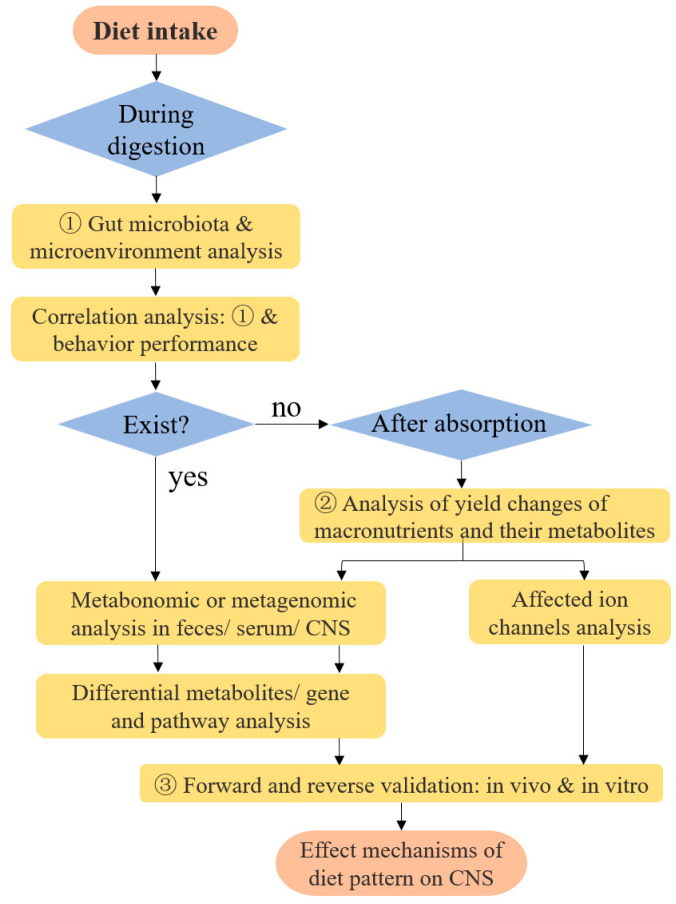
Referable research models of how dietary patterns affect CNS functions.

**Figure 5 nutrients-15-05080-f005:**
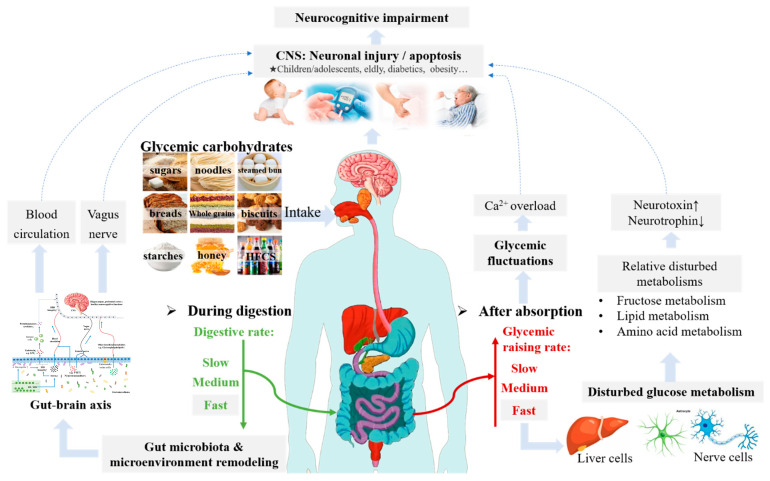
A summary of the effect mechanisms of glycemic carbohydrate diet on neurocognitive.

**Table 1 nutrients-15-05080-t001:** Studies for several diet-induced metabolic diseases and their impairments of neurocognitive functions.

Research Model		Diet-Induced Diseases	Control	Behavioral Tests	Outcomes and Conclusions	Ref.
HUMANS	Diabetics: 40–75 years old	T2DM	People with normal glucose metabolism	Tests in 3 cognitive domains: memory/attention/information processing speed	Diabetics performed worse in all cognitive domains. It can be largely explained by hyperglycemia.	[8]
	Diabetics: 70.9 years old	T2DM with sever glucose variability	T2DM with relatively stable blood glucose	Annually observational follow-up for 4.8 years	Cognitive functions can be influenced by glucose variability independently of mean blood glucose.	[7]
	Young adults: 20 years old	Obesity, prediabetes	Persons with normal glucose	Inhibitory control/sustained attention/working memory	Higher glucose levels were associated with poorer cognitive performance, especially for prediabetes.	[12]
MICE	8 weeks old	T2DM induced by STZ and 20% Fr solution	Healthy mice with standard chow and water	After 38 days intervention	There is a strong association between hyperglycemia, hyperinsulinemia, neuroinflammation, and cognitive dysfunction in T2DM mice model.	[51]
	4 months of age	T2DM (db/db mice): ad libitum to standard chow	db/db mice: Intermittent fasting (IF)	After 28 days of exposure	db/db: cognitive decline; db/db-IF:cognitive improved; mitochondrial biogenesis and energy metabolism gene expression in hippocampus increased; microbial metabolites re-structured.	[44]
	Juvenile: 5 weeks old; elderly: 1 year old	Obesity induced by HFD	Healthy mice with standard chow	After 11/24 weeks of exposure	Obesity causes a dysmetabolic phenotype in both age groups. Older age exacerbates neuroinflammatory response and cognitive decline.	[36]
	8 weeks old	Obesity induced by HFD and 34% Su solution	Healthy mice with standard chow and water	After 10 weeks feeding of high-calorie diet (HFD/HSD)	HFD/HSD intervention produces obesity and cognitive decline, which is accompanied by increased microglial activation and reduced numbers of dendritic spines.	[49]
	8 weeks old	Hyperglycemia induced by chronic social defeat (CSD)	Healthy mice with normal blood glucose	3- and 5 weeks post-CSD	Hyperglycemia threatens long-term glucose homeostasis and causes spatial memory dysfunction.	[52]
RATS	5 weeks old	Hyperglycemia induced by STZ (STZ group)	Blood glucose controlled with insulin injection (STZ + insulin group)	60 days after blood glucose control	Chronic hyperglycemia can compromise cognition by reducing hippocampal ERK signaling and inducing neurotoxicity.	[53]
	Rat pups	Hyperglycemia induced by STZ /glucose injection	Treated with equal citrate buffer	After 10 days of glucose injection/5 days of STZ treatment	Hyperglycemia alters substrate transport, lactate homeostasis, dendritogenesis, and glutamate—glutamine cycling in the developing hippocampus.	[54]

Notes: T2DM, type 2 diabetes; STZ, streptozotocin; HFD, high-fat diet; ERK: extracellular regulated protein kinases.

**Table 2 nutrients-15-05080-t002:** Studies for different types of GCs intervention-induced changes in neurocognitive functions.

Research Model		Dietary Patterns		Control	Exposure	Behavioral Tests	Outcomes and Conclusions	Ref.
HUMANS	Women: 50–79 years old	HSD	Usual diet (higher dietary sugar intake)	Behavioral modification training	8.1 years	Annually observational follow-up for 15 years	An estimated increase of 10 g/day in total sugar intake was associated with an increased AD risk by 1.3–1.4%.	[39]
	Young adults: about 23 years of age	NNS	Fr/Gl/Su solution	Sucralose solution	Instant testing	After 250 mL solution drink	Gl and Su led to poorer performance on the assessed tasks as opposed to Fr and placebo, especially under the fasting condition.	[160]
			(equal sweetness intensity)					
	Healthy elderly (72.9 years old) participants	Whole grains	High hydrostatic pressurizing brown rice (UHHPBR)	Polished white rice (WR)	24 months	After 24 months of exposure	Long-term consumption of UHHPBR increases information processing speed in the elderly, suggesting a protective effect of UHHPBR administration against age-related cognitive decline.	[161]
MICE	Juvenile: 4 weeks old	Fr	High Fr diet (30% calories)	Standard chow	12 weeks	—	High Fr feeding leads to damaged IEB, elevated serum endotoxin levels, hippocampal neuroinflammatory response, and neuronal loss.	[76]
	11 weeks old	Whole grains	Oat β-glucan added in HFD	HFD; Standard chow	15 weeks	After 15 weeks of exposure	β-glucan intake can improve gut barrier function, reduce endotoxemia, and enhance cognitive function via more optimized synaptic and signaling pathways in critical brain areas.	[162]
	18 weeks old		BWF	Standard chow	15 weeks	After 26 weeks of exposure	BWF intake can suppresses cognitive decline by increasing hippocampal BDNF production in SAMP8 mice.	[163]
RATS	Adolescents: PN 21; young adults: PN 56	Su	10% Su solution	0.1% sodium saccharin solution (standard chow)	4 weeks	Adolescents: PN 55; young adults: PN 91	Sucrose intervention can disrupt spatial cognition and reward-related behavior in the absence of obesity.	[164]
	Adults: age not specified		10% Su solution	Water (standard chow)	3 weeks	After 21 days of exposure	Sucrose intervention can disrupt hippocampal-dependent place recognition memory; neuroinflammation and OS play a role in this impairment.	[165]
	Adolescents: PN 28		10% Su solution	Water (standard chow)	5 weeks	PN 62	The expression of parvalbumin-immunoreactive GABAergic interneurons has decreased; both prefrontal and hippocampal functions have declined.	[166]
	8 weeks old	Fr	10% Fr solution	Water (standard chow)	8 months	After 8 months of exposure	High Fr diet induced peripheral IR and an abnormal insulin-signaling pathway in the hippocampus, which exacerbated memory deficits.	[167]
	6 weeks old		15% Fr solution	Water (standard chow)	24 weeks	7, 10, 14, 16, 18, 20, 22, and 24 weeks	IR/cognitive dysfunction appeared from 7th/20th week. Fr-induced neuroinflammation and OS impaired neuronal signaling and synaptic plasticity.	[168]
	8 weeks old		10% Fr solution	Water (standard chow)	7 months	After 7 months of exposure	The induced cognitive deficits are related to increased OS, hypertriglyceridemia, impaired insulin signaling, and altered mitochondrial dynamics.	[169]
	Mother rats: from GD0 (gestational day)		13%/40% Fr solution	Water (standard chow)	GD0-PN 21 (offspring)	Postnatal day 60 offspring	Maternal Fr exposure during gestation and lactation can impair cognition in offspring and affect brain function at the transcriptome level.	[170]
	Adolescents: PN 30 /Adults: PN 60	HFCS	11% Su solution /11% HFCS-55	Water (Low-fat chow)	4 weeks	PN 60 /PN 90	Adolescents: both learning and memory functions have declined; adults: no significant impact.	[34]
	Juvenile /Adolescent rats: PN 26		HFCS-55	Water (standard chow)	4 weeks	PN 175	HSD in early life may confer long-lasting impairments in memory function, which are not reversible by simply removing sugars from the diet.	[31]
	Juvenile/ adolescents: PN 26–28		65% Fr + 35% Gl soluton	Water (standard chow)	6 weeks	PN 67	The abundance of P. distasonis and P. johnsonii has elevated; hippocampal function has declined.	[33]
	Healthy/T2DM adult rats	NNS	Aspartame (ASP) solution	0.9% NaCl	30 days	After 30 days of oral gavage	ASP administration to healthy/diabetic rats has shown adverse effects linked to cognitive dysfunction.	[171]
	Adolescents: PN 25		Acesulfame potassium/saccharin/stevia (LCS) solution	Water (standard chow)	30 days	After 30 days of exposure	Habitual-life LCS consumption has long-lasting implications for hippocampal-dependent memory in rats.	[172]
	6–7 weeks old		Su/saccharin solution	Water (standard chow)	10% Su: 4 weeks; Su/water/saccharin: 4 weeks	After 4–8 weeks of exposure	4 weeks of Su exposure results in cognitive decline. Switching from Su to water or saccharin produces similar improvements on cognitive measures.	[173]
	8–11 weeks old	Maltodex -trin	10.4% Su/maltodextrin solution	Water (standard chow)	17 days	After 17 days of exposure	Impaired performance on a location recognition task was found in both groups.	[112]

Notes: Fr: fructose; Gl: glucose; Su: sucrose; HFCS-55: high fructose corn syrup; PN: postnatal day; HSD, high-sugar diet; IEB: intestinal epithelial barrier; IR, insulin resistance; BDNF, brain-derived neurotrophic factor; NNS: non-nutritive artificial sweeteners.

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
