# Peer review of "A Comprehensive Review of the Effects of Glycemic Carbohydrates on the Neurocognitive Functions Based on Gut Microenvironment Regulation and Glycemic Fluctuation Control"

_nutrients, 2023, doi:10.3390/nu15245080_

Round 1

Reviewer 1 Report

Comments and Suggestions for Authors

The authors have written an interesting review of the literature describing the association between glucose/readily digestible carbohydrates and cognitive function. The review is thorough in its explanation of possible mechanisms and the role of the intestinal microbiota.

Comments:

1) Despite certification of English editing, the manuscript contains a large number of errors. Repeat editing is required. For example:

Line 17: prone is an incorrect term. Prefer "which may lead to cognitive impairment."

39: "It" should be "They mainly consist.."

44: "fat" is not a disease.

79, 81: lower-case M for monosaccharides & malto-oligo...

91: "that" change to "toward incrementally.. become"

98: define GL

119: "instant" change to acute

132, 133: Throughout the manuscript text (the tables show the species), the authors fail to specify the animal model. When "childhood/adolescence" is stated, this usually assumes human. Be clear about the animal used in the studies cited.

141: "older" what?

175: "researches" is not correct.

Table 1: suggest "research model" instead of "object"

183: "reduced gut microbiota" - do you mean reduced diversity?

186: "increased" gut barrier integrity - do you mean "impaired"?

216: be transported

221: what is "appropriate"?

227: the gut microbiota

231: "decreased" gut microbiota?

259: as stated above, species/model must be specified. Bumblebees are far removed from humans.

266: 20% of the oxygen...25% of the glucose...

274, 517: "metabolisms" is not correct.

306-314 & elsewhere: the bolded letters within a paragraph are not proper English. Use indentations or "First..." "Second..."

Do you reference Figure 1 A in the text?

538: hyperglycemia is not a disease

540: change "maleficant" to pathogenic.

544: change "detected" to associated with

561: why do you call it "subtle"?

579: remove "once"

620: why do you state that children have "weak" gut function?

629: Specify the species.

664, 665: These are incomplete sentences.

667: study models

668: clarify what you mean in stating that dietary interventions are "mild"; why are they time-consuming?

677, 678: does modern society really exhibit more mental labor and desire for immediate happiness compared to previous societies?

681: specify "different" - profiles?

682: sentence is incorrect

331: Figure 1 (B)

347, 351: use upper-case A B, as you do in the figure.

354: sentence needs correction.

384: "decomposed" change to converted or metabolized.

390: higher than what?

418: low-grade inflammation

462: remove "in their own right."

472: define Orai

Comments on the Quality of English Language

Extensive editing is still needed.

Reviewer 2 Report

Comments and Suggestions for Authors

This great review discusses an ill-known cognitive feature of carbohydrate overload (which becomes even more clear in diabetes). The review is, indeed, comprehensive, structured and informative. I have some remarks:

- Section 1-2: The side-effects of the ketogenic diet are treatable and not too dangerous under specialized supervision, which should be noted by the authors. It is still a valuable tool in epilepsy, for instance.

- Table 1-2: Please do group each species apart, as not all results in mice and/or rats can be readily extrapolated to patients.

Comments on the Quality of English Language

Just some minor grammatical errors throughout the text.

Round 2

Reviewer 1 Report

Comments and Suggestions for Authors

The authors have made the appropriate revisions

One additional correction to complete:

Table 2 NNS needs defining in the footnote.

Author Response

Thank you for your suggestion.  We have defined NNS, which stands for non-nutritive artificial sweeteners, in the footnote of Table 2.
